# Node-oriented Spectral Filtering for Graph Neural Networks

## Abstract

Graph neural networks (GNNs) have shown remarkable performance on homophilic graph data while being far less impressive when handling non-homophilic graph data due to the inherent low-pass filtering property of GNNs. In general, since the real-world graphs are often a complex mixture of diverse subgraph patterns, learning a universal spectral filter on the graph from the global perspective as in most current works may still be difficult to adapt to the variation of local patterns. On the basis of the theoretical analysis of local patterns, we rethink the existing spectral filtering methods and propose the Node-oriented spectral Filtering for Graph Neural Network (namely NFGNN). By estimating the node-oriented spectral filter for each node, NFGNN is provided with the capability of precise local node positioning via the generalized translated operator, thus adaptive discriminating the variations of local homophily patterns. Furthermore, the utilization of re-parameterization brings a trade-off between global consistency and local sensibility for learning the node-oriented spectral filters. Meanwhile, we theoretically analyze the localization property of NFGNN, demonstrating that the signal after adaptive filtering is still positioned around the corresponding node. Extensive experimental results demonstrate that the proposed NFGNN achieves more favorable performance.

## 1   Introduction

As a powerful tool for analyzing graph data, GNNs are attracting considerable attention from both academia and industry. Meanwhile, GNNs have also demonstrated remarkable capabilities in a number of graph-related applications, including but not limited to recommendation system [14, 40], disease prediction [27, 17], drug discovery [36, 9], and action recognition [42, 33].

In the field of graph machine learning, homophily has always remained a common assumption [25, 39], i.e., nodes within the same class tend to connect with each other. However, behind the great success of the previous efforts, such assumption as a critical limitation doesn't hold true in many graph-related scenario, which inhibits severely the further extension of GNNs to more general graph data. As a mater of fact, it is hard to argue that homophily is an inherent characteristic of graph data [45] and there are also a considerable number of non-homophilic graphs in real-world, where the links usually exist between nodes from different classes. For example, in protein structural networks, connections between different types of amino acids are easier to form [8]; in addition, for an air traffic network, the establishment of the air routes is more for commercial reasons and has little to do with the activities of airports [29].

As far as the existing GNNs are concerned, most of them usually adopt message passing architecture in the spatial domain to aggregate the node feature from neighbors over the given topology structure [38, 10, 20]. Obviously, the practice that all neighbor nodes are considered to contribute positively to

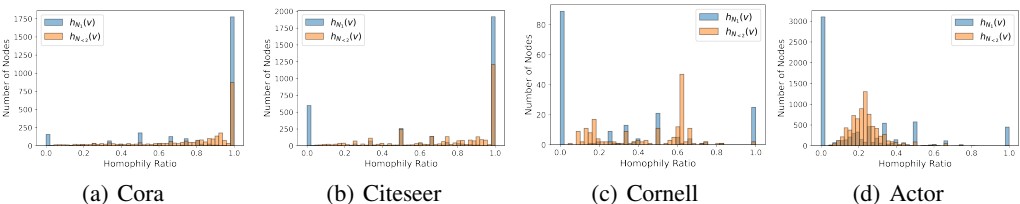

|        |         |         |        |
|--------|---------|---------|--------|
| (a) Cora | (b) Citeseer | (c) Cornell | (d) Actor |

Figure 1: The statistical histogram of $h_{N_1}(v)$ and $h_{N_{<2}}(v)$ of four real-world graphs, where Cora and Citeseer are known as the graphs with strong homophily, Cornell and Actor are known as the graphs with strong heterophily.

node aggregation without distinction does not apply to heterogeneous graphs. Besides, the commonly adopted message passing architectures have been proven to exhibit significant low-pass filtering properties [2, 41], which is quite contrary to the non-low-pass properties of non-homophilic graphs. In order to break the homophily limitation, several recent studies have made some exploratory efforts to solve the GNN modeling problem with non-homophily graphs, such as exploring some new aggregation schemes [45, 12, 44] and high-pass spectral filtering [4, 46]. However, these methods are still designed for specific heterophilic graph and lack of good scalability to homophilic graph. Actually, whether a graph is homophilic or heterophilic depends on the relatedness of the downstream task to the graph construction. It means that for a fixed topology structure, when combined with different downstream tasks, the identification of its graph property will be very different. Taking the dating network as an example, people are more likely to date someone who are opposite sex and about the same age. In this case, the graph is likely to be heterophilic if we use the gender of the node as the label, while it may be homophilic if we use the age group of the node as the label. Therefore, from the perspective of practical application, a simple yet effective GNN model that can be adaptively applied to graphs with mixed structural properties should be more preferred.

Moreover, GNN also confronts with another follow-on challenge, i.e., the homophily property is in general not consistent across the whole graph. In a real-world graph, there are always diverse subgraph patterns among different regions [37]. As shown in Fig 1, a network seems like homophily may also contain a small amount of randomness or heterophily. Although the universality of GNNs has been taken into consideration in [6, 13] through adaptive spectral filter learning, the global filter modeling without focus on the variation of local structural pattern may still be suboptimal for the graph that is mixed up of more complex homophilic and heterophilic property. Besides, the relation re-estimation based methods [24, 37, 16, 28] have shown some advance in addressing the issue of the mixing pattern to a certain extent. In these approaches, different measures of node similarity are defined to perform potential neighborhood discovery, whereas the design of similarity measure and the high complexity that comes with it makes them less concise and flexible.

In this paper, we first analyze the local mixing patterns in the graph via the label consistency of the node neighborhoods. A theoretical justification is also given to analyze why the existing near-neighborhood aggregation mechanisms fail to work for the non-homophily graphs. Further, inspired by the generalized translated operator in graph signal processing (GSP), we propose a novel GNN (namely NFGNN) from the perspective of spectral filtering to achieve adaptive localized graph spectral filter learning. The key idea is to estimate the node-oriented filter for each node to solve the issue of varied local patterns in the graph. To be specific, we first apply the translated operator to center the spectral filter at each node, and then the K-order polynomial is used to approximate the optimal filter to be learned at each node. In addition, low-rank approximation based re-parameterization is used to decompose the filter weight matrix to node-agnostic and node-dependent matrices, improving the flexibility of the model. It also brings a trade-off between global and local perspectives. Meanwhile, we theoretically prove that the filtered signal is localized around the corresponding node, demonstrating that NFGNN achieves the adaptive localized filtering. Finally, an extensive group of experiments on various real-world datasets with different scales verifies that the proposed NFGNN achieves more favorable performance.

## 2 Preliminaries

**Notations.** An undirected graph is denoted as $\mathcal{G} = (\mathcal{V}, \mathcal{E})$, where $\mathcal{V} = \{v_i\}_{i=1}^{|\mathcal{V}|}$ denotes the set of nodes with $|\mathcal{V}| = n$, and $\mathcal{E}$ is the set of edges among nodes. The topology structure of graph $\mathcal{G}$

81 could be described by the adjacency matrix $\mathbf{A} \in \mathbb{R}^{n \times n}$ with $\mathbf{A}_{i,j} = 1$ if $(i, j) \in \mathcal{E}$ or 0 otherwise,
82 and $\mathbf{D}$ is the diagonal degree matrix $\mathbf{D}$ with its $i$-th diagonal entry $\mathbf{D}_{ii} = \sum_j A_{ij}$. Besides, we use
83 $\mathbf{L} = \mathbf{I} - \mathbf{D}^{-1/2} \mathbf{A} \mathbf{D}^{-1/2}$ to denote the symmetric normalized Laplacian matrix of $\mathcal{G}$ and $\mathbf{I}$ is the
84 identity matrix.

85 For each node $v \in \mathcal{V}$, we denote its neighborhood by using $N(v)$, and further, the $i$-hop neighbors
86 $N_i(v)$ and the neighbors within $i$-hops $N_{<i}(v)$ of node $v$ by $N_i(v) = \{m : m \in \mathcal{V} \wedge d_{\mathcal{G}}(v, m) = i\}$
87 and $N_{<i}(v) = \{m : m \in \mathcal{V} \wedge d_{\mathcal{G}}(v, m) \leq i\}$, respectively, where $d_{\mathcal{G}}(i, j)$ is the shortest path
88 distance between two nodes $i$ and $j$ on graph $\mathcal{G}$. Besides, let $\mathbf{x} = [x_1, \cdots, x_i, \cdots, x_n]^T \in \mathbb{R}^n$
89 denote the $n$-dimensional signal defined on the given graph $\mathcal{G}$, where $x_i$ denotes the signal response
90 (feature) at the $i$-th node $v_i$. Generally, when each node receives $f$ channels of signals, we will have
91 a feature matrix $\mathbf{X} = [\mathbf{X}_1, \cdots, \mathbf{X}_i, \cdots, \mathbf{X}_n]^T \in \mathbb{R}^{n \times f}$ with each column of it being a graph signal
92 $\mathbf{x}$ and its $i$-th row $\mathbf{X}_i \in \mathbb{R}^f$ representing $f$- dimensional feature vector associated with node $v_i$[1].

93 Furthermore, for the node classification task, each node $v \in \mathcal{V}$ has a class label $y_v \in \mathcal{Y} = \{1, \cdots, C\}$,
94 where $\mathcal{Y}$ is the set of class labels with $|\mathcal{Y}| = C$, and $C$ is the number of classes. In addition, we use
95 $\mathbf{y}_v \in \mathbb{R}^C$ to denote the one-hot vector corresponding to $y_v$.

96 **Graph Fourier Transform.** According to graph signal processing theory, the graph Laplacian
97 provides an effective way of spectral analysis on graphs. Given the Laplacian matrix $\mathbf{L}$, it can be
98 eigendecomposed into $\mathbf{U}\Lambda\mathbf{U}^T$, where $\mathbf{U} = [\mathbf{u}_1, \cdots, \mathbf{u}_l, \cdots, \mathbf{u}_n] \in \mathbb{R}^{n \times n}$ is the graph Fourier basis
99 formed by $n$ orthonormal eigenvectors $\{\mathbf{u}_l\}_{l=1}^n$, and $\Lambda = \mathrm{diag}[\lambda_1, \cdots, \lambda_l, \cdots, \lambda_n] \in \mathbb{R}^{n \times n}$ is the
100 diagonal matrix of the ordered eigenvalues $\{\lambda_l\}_{l=1}^n$, $\lambda_l \in [0, 2]$. Notice that $\{\lambda_l\}_{l=1}^n$ is also identified
101 as the frequencies of the graph. Thus, the graph Fourier transform of the signal $\mathbf{x}$ is defined as
102 $\hat{\mathbf{x}} = \mathbf{U}^T \mathbf{x}$, and $\hat{x}(\lambda_l)$ indicates the response of $\mathbf{x}$ at the frequency $\lambda_l$. The inverse graph Fourier
103 transform is defined as $\mathbf{x} = \mathbf{U}\hat{\mathbf{x}}$ [34]. Thus, on the basis of the graph Fourier transform, the signal $\mathbf{x}$
104 filtered by $\hat{g}$ can be given as follows:

$$\mathbf{z} = \sum_{l=1}^n \hat{g}(\lambda_l) \mathbf{u}_l \mathbf{u}_l^T \mathbf{x} = \mathbf{U}\hat{\mathbf{g}}\mathbf{U}^T \mathbf{x} \tag{1}$$

105 where $\hat{\mathbf{g}} = \hat{g}(\Lambda) = \mathrm{diag}[\hat{g}(\lambda_1), \cdots, \hat{g}(\lambda_l), \cdots, \hat{g}(\lambda_n)]$ is the spectral filter and we have $\mathbf{g} = \mathbf{U}\hat{\mathbf{g}}$.
106 Since the spectral filtering is equivalent to convolution in the spatial domain [26], Eq. 1 could also
107 be defined as the spectral graph convolution $\mathbf{z} = \mathbf{x} *_{\mathcal{G}} \mathbf{g}$, where $*_{\mathcal{G}}$ denotes the graph convolution
108 operator.

## 3 Motivations

### 3.1 Adaptability to Mixing Local Structural Patterns

111 To measure the homophily of a graph, both edge homophily ratio [45] and node homophily ratio [28]
112 are two widely used metrics. In addition, [23] proposed a more comprehensive homophily metric that
113 mitigates homogeneity bias from class imbalance. It is less sensitive to the number of classes and size
114 of each class than edge homophily ratio and node homophily ratio. Since we aim to analyze the local
115 patterns of the graph via the label consistency of the node neighborhoods, the node homophily ratio
116 is adopted in this work. It should be noticed that the edge homophily ratio and the node homophily
117 ratio have similar qualitative behavior [21]. In particular, the node homophily ratio $\mathcal{H}_{\mathcal{G}}$ of the graph
118 $\mathcal{G}$ is defined as the average of the homophilic 1-hop neighbor ratio of each node $v$ in $\mathcal{G}$ and given by:

$$\mathcal{H}_{\mathcal{G}} = \frac{1}{|\mathcal{V}|} \sum_{v \in \mathcal{V}} h_{N_1}(v) = \frac{1}{|\mathcal{V}|} \sum_{v \in \mathcal{V}} \frac{|\{m \in N_1(v) : y_m = y_v\}|}{|N_1(v)|} \tag{2}$$

119 where $h_{N_1}(v)$ denotes the homophilic 1-hop neighbor ratio of node $v$ centered at node v.

120 In essence, $H_{\mathcal{G}}$ provides an overall evaluation criterion for homophily of graph. Instead, we should
121 also cast more insight into the variation of local structural pattern. Particularly, we first give vi-
122 sualization of the statistical histogram of $h_{N_1}(v)$ and $h_{N_{<2}}(v)$. As shown in Fig 1, even in Cora

---

[1]Unless otherwise stated, only $\mathbf{x} \in \mathbb{R}^n$ is considered as the input of GNNs for convenience of presentation,
the following discussions of this work still apply to $\mathbf{X} \in \mathbb{R}^{n \times f}$ with $f$ channels of signals (i.e., $f$ -dimensional
features).

and Citeseer network, which are usually considered as homophilic graphs, there still exist a small number of completely 1-hop heterophilic subgraphs. Similarly, there are some subgraphs with a high homophily ratio in Cornell and Actor network. Obviously, these observations mean that, for a graph with complex topological structure, it is definitely a mixture of homophilic and heterophilic local subgraphs. Furthermore, for the two heterophily graphs, we can find that the statistical histogram of $h_{N_{<2}}(v)$ is much different from that of $h_{N_1}(v)$, demonstrating that the associated local subgraph patterns for each node varied generally with the change of neighborhood range.

Based on the above analysis on the variation of local structural pattern, it motivates us to consider that it is conducting adaptive modeling for the graph nodes with different degree of homophily is a necessity, and further, we should improve the effectiveness of GNNs for the nodes with various local structural patterns.

## 3.2 Aggregatability of Near-neighbors

To facilitate the discussion of the aggregatability of near-neighbors, we first give two definitions about the neighborhood, i.e., *heterophily-preferred* and *homophily-preferred*:

**Definition 3.1.** *For a node $v$ with label $y_v$, $N(v)$ is expected to be heterophily-preferred if $P(y_m = y_v|y_v) \leq P(y_m \neq y_v|y_v)$, $\forall m \in N(v)$. Conversely, $N(v)$ is expected to be homophily-preferred.*

Intuitively, the near-neighbor aggregation is definitely effective when the near-neighbor subgraph is completely homophilic, while it may not capture adequate homophilic information when the neighborhood is expectedly heterophily-preferred. According to Definition 3.1, it can be inferred that the aggregation of expectedly homophily-preferred neighborhoods is also beneficial to the node representation. Classical GNNs [19, 38, 10] commonly suffer from the over-smoothing problem and are thus limited to shallow networks, which means that each node just aggregates the information about its neighbors within 2 or 3-hops. Thus, whether the near-neighborhood is homophily-preferred or heterophily-preferred will be of great importance for them. To empirically analyze the preference of near-neighborhood, we first propose a label entropy $S_{N_i}(v)$ to measure the neighbor label distribution of node $v$, which is defined as :

$$S_{N_i}(v) = -\sum_{y\in\mathcal{Y}}(\frac{|N_i^{(y)}(v)|}{|N_i(v)|} + \varepsilon)\log(\frac{|N_i^{(y)}(v)|}{|N_i(v)|} + \varepsilon) \tag{3}$$

where $N_i^{(y)}(v) = \{m : m \in N_i(v) \wedge y_m = y\}$ and $\varepsilon =$1e-10 is a constant to avoid overflow. Clearly, the larger the label entropy $S_{N_i}(v)$ is, the more random the neighbor label distribution of $v$ will be. As shown in Fig 2, most nodes in the homophily graphs have low $S_{N_1}(v)$ and most nodes in the heterophily graphs have high $S_{N_1}(v)$. Besides, for all four graphs, the statistical histogram of $S_{N_{<2}}(v)$ is shifted to the right overall compared to $S_{N_1}(v)$. These observations suggest that the neighbor label distribution of each node tends to be uniform as the neighborhood range increases. Combined with the definition 3.1, we can conclude that the near-neighbor based aggregation is not the optimal solution for heterophily graphs.

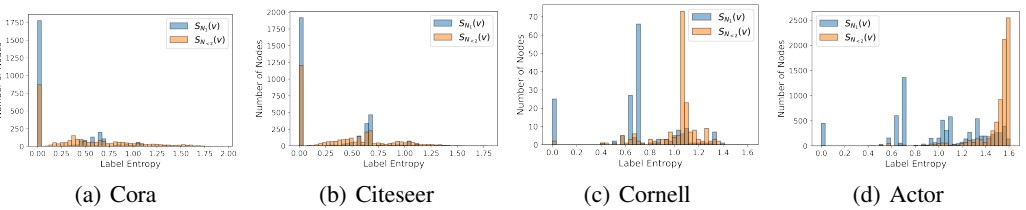

(a) Cora      (b) Citeseer      (c) Cornell      (d) Actor

Figure 2: The statistical histogram of $S_{N_1}(v)$ and $S_{N_{<2}}(v)$ of four real-world graphs.

Theoretically, we explore the preference of the 2-hop neighborhood $N_2(v)$ for multi-class node classification and have the following proposition:

**Proposition 3.1.** *For each node $v$ in a graph $\mathcal{G}$, let's assume the class labels of its neighbors $\{y_m : m \in N(v)\}$ are conditionally independent when given $y_v$, and $P(y_m = y_v|y_v) = \alpha$, $P(y_m = y|y_v) = \frac{1-\alpha}{|\mathcal{Y}|-1}, \forall y \neq y_v$. Then, the 2-hop neighborhood $N_2(v)$ of a node $v$ will always be expectedly heterophily-preferred if $\alpha \leq \frac{2}{|\mathcal{Y}|}$.*

163  For the proof of the above proposition 3.1, please refer to the supplemental materials.

164  Different from the statement that 2-hop neighbor aggregation could help GNNs to learn more
165  effective node representation [45, 1], proposition 3.1 shows that the aggregation of $N_2(v)$ is hard to
166  be beneficial for GNNs when the neighborhood label distribution tends to be uniform. This finding is
167  also consistent with our empirical analysis mentioned above.

## 4  The Proposed NFGNN

169  The observations in Sect. 3 points out that the real-world graph is often made up of a mixture of various
170  local patterns, while the near-neighbor aggregation mechanism does not handle the heterophilic local
171  patterns well. Various elaborately designed aggregation mechanisms in the spatial domain have been
172  proposed to tackle these issues. Different from these methods based on spatial domain, spectral graph
173  convolution aims to learn a specific spectral filter for a given graph structure and node labels, thus
174  preserving the appropriate frequency components for the downsteam tasks. Therefore, spectral graph
175  convolution also possesses a strong expressive power [2] and can work well for both heterophily and
176  homophily graphs. However, existing spectral-based methods are still not flexible enough. They
177  usually estimate a globally consistent filter from the perspective of the whole graph [20, 13, 7, 3],
178  which may be inappropriate for some local patterns. In this section, we rethink the globally consistent
179  spectral graph convolutions, and propose a localized spectral filter learning method to break the
180  limitation.

### 4.1  Polynomial Filter

182  Based on Eq 1, early spectral GNNs [5, 15] directly eigendecompose the normalized Laplacian
183  matrix $\mathbf{L}$ to obtain the Fourier basis $\mathbf{U}$ and treat the $\hat{g}$ as the trainable parameters. However, the
184  expensive eigenvalue decomposition restricts the availability of these methods greatly. To circumvent
185  the eigendecomposition, $K$-order polynomial approximation is adopted to parameterize the spectral
186  filter:

$$\hat{g}(\lambda_l) \approx \sum_{k=0}^{K} \gamma_k \lambda_l^k \tag{4}$$

187  where $\gamma_k$ denotes the learnable fitting coefficient. By plugging Eq. 6 into Eq. 1, the spectral filtering
188  can be rewritten as:

$$\mathbf{U}\hat{\mathbf{g}}\mathbf{U}^T\mathbf{x} \approx \mathbf{U}\left(\sum_{k=0}^{K} \gamma_k \Lambda^k\right)\mathbf{U}^T\mathbf{x} = \left(\sum_{k=0}^{K} \gamma_k \mathbf{L}^k\right)\mathbf{x} \tag{5}$$

189  Except for the reduced complexity, another advantage of the polynomial-parameterized filter is the
190  localization property. When the filter $\hat{\mathbf{g}}$ centers at node $v_i$, the value at node $v_j$ after filtering by $\hat{\mathbf{g}}$ is
191  equal to $\sum_{k=0}^{K} \gamma_k (\mathbf{L}^k)_{i,j}$. Meanwhile, $(\mathbf{L}^k)_{i,j}$ will be 0 if $d_{\mathcal{G}}(i,j) > K$ [11]. The above facts show
192  that the $K$-order polynomial spectral filter is exactly localized in $N_{<K}(i)$.

193  Due to the high efficiency, various polynomial kernels are used for spectral filter parameterization,
194  such as Chebyshev basis [7] and Bernstein basis [13]. Interestingly, many spatial aggregation
195  methods can also be essentially attributed to polynomial-parameterized spectral convolution [20, 3].
196  Nevertheless, such polynomial filters are still globally consistent, or node independent.[2] In other
197  words, the filter $\hat{\mathbf{g}}$ is applied for all nodes with the fixed fitting coefficients $\{\gamma_k\}_{k=0}^{K}$ that are trained
198  on the whole graph, and makes no specific discrimination for each node when performing filtering.
199  Thus, even it is localized, the polynomial-parameterized spectral filters are still unable to effectively
200  model the complex local structural patterns with a mixture of homophily and heterophily. Intuitively,
201  compared to learning a globally shared filter $\hat{g}(\lambda_l)$ as a trade-off solution for different local patterns
202  across the whole graph, learning an appropriate node-specific filter $\hat{g}_i(\lambda_l)$ for node $i$ to fit the local
203  pattern where it is located seems to be a better choice.

$$\hat{g}_i(\lambda_l) \approx \sum_{k=0}^{K} \gamma_{i,k} \lambda_l^k \tag{6}$$

204  For such practice, what needs to be figured out is how to learn $\hat{g}_i$ and ensure it is still positioned
205  around node $i$. To this end, we introduce the node-oriented filtering.

---

[2]The analysis of existing GNNs from a spectral filtering perspective is provided in the supplemental materials.

## 4.2 Translated Filter for Node-oriented Filtering

Inspired by the generalized translation operator, we develop an adaptive localized spectral filtering on graph $\mathcal{G}$ using the polynomial-parameterized spectral convolution. It takes full into account the specific effect of the node where the filter is positioned.

**Definition 4.1.** ( *Generalized translation operator*) *[34] For any signal* $\mathbf{g} \in \mathbb{R}^n$ *defined on a given graph* $\mathcal{G}$ *and any* $i \in \{0, 1, \cdots, n-1\}$, *we define a generalized translation operator* $\mathbf{T}_i : \mathbb{R}^n \to \mathbb{R}^n$ *via generalized convolution with a Kronecker delta function* $\delta_i$ *centered at the $i$-th node $v_i$:*

$$\mathbf{T}_i(\mathbf{g}) := \sqrt{N}(\mathbf{g} * \delta_i) = \sqrt{N} \sum_{l=1}^{n} \mathbf{u}_l u_l^T(i) \hat{g}(\lambda_l) \tag{7}$$

*where* $u_l^T(i)$ *denotes the $i$-th element of* $\mathbf{u}_l^T$.

Definition 4.1 shows that a signal could be centered at a specific node through a kernelized operator acting on $\hat{\mathbf{g}}$ [34]. Then, we can perform the spectral convolution of signal $\mathbf{x}$ with the filter signal $\mathbf{g}$ when $\mathbf{g}$ is centered at a specific node $v_i$:

$$\mathbf{x} *_{\mathcal{G}} \mathbf{T}_i(\mathbf{g}) = \sqrt{N} \sum_{l=1}^{n} \mathbf{u}_l \hat{x}(\lambda_l) u_l^T(i) \hat{g}(\lambda_l) \tag{8}$$

Let $\hat{g}_i(\lambda_l) = \sqrt{N} u_l^T(i) \hat{g}(\lambda_l)$, Eq. 8 becomes:

$$\mathbf{x} *_{\mathcal{G}} \mathbf{T}_i(\mathbf{g}) = \sum_{l=1}^{n} \hat{g}_i(\lambda_l) \mathbf{u}_l \mathbf{u}_l^T \mathbf{x} = \mathbf{U} \hat{\mathbf{g}}_i \mathbf{U}^T \mathbf{x} \tag{9}$$

Recall the inverse graph Fourier transform, it can be derived that $x_i = \mathbf{U}_{i:} \hat{\mathbf{x}}$, where $\mathbf{U}_{i:}$ indicates the $i$-th row of $\mathbf{U}$. Hence, $\mathbf{U}_{i:}$ could also approximated from $x_i$ according to $\mathbf{U}_{i:} \approx x_i \mathbf{q}$, where $\mathbf{q} = \text{pinv}(\hat{\mathbf{x}}) \in \mathbb{R}^n$ is the pseudoinverse of $\hat{\mathbf{x}}$. Then, note that $u_l(i) = u_l^T(i)$, it can be derived that $\hat{g}_i(\lambda_l) \approx \tilde{g}_i(\lambda) = \sqrt{N}(x_i q_l) \hat{g}(\lambda_l)$, where $q_l$ denotes the $l$-th element of $\mathbf{q}$. Therefore, $\hat{\mathbf{g}}_i$ can be approximated by a specific filter $\tilde{\mathbf{g}}_i$ corresponding to node $v_i$, which also considers the impact from the feature $x_i$ associated with node $v_i$ in estimating the filter weight.

Without loss of generality, the $K$-order polynomial approximation can be used to directly parameterize $\tilde{\mathbf{g}}_i$. As a traditionally used approximate kernel in GSP [7], the Chebyshev polynomial $T_k(\cdot)$ are adopted to parameterize $\tilde{\mathbf{g}}_i$, that is, $\tilde{\mathbf{g}}_i = \sum_{k=0}^{K} \eta_{i,k} T_k(\tilde{\Lambda})$, where $\tilde{\Lambda} = 2\Lambda/\lambda_{max} - \mathbf{I}$. Meanwhile, for each node, we focus only on the convolution result of the filter positioned at that node, i.e., $\mathbf{z}_i = \delta_i(\mathbf{U}\tilde{\mathbf{g}}_i\mathbf{U}^T\mathbf{x})$, here $\delta_i = [0, \cdots, \underset{i}{1}, \cdots, 0] \in \mathbb{R}^n$ denotes a row vector with only the $i$-th element being 1 and the remains being zeros. Thus, the node-oriented localized filtering can be as:

$$\mathbf{z}_i = \delta_i(\mathbf{U}\tilde{\mathbf{g}}_i\mathbf{U}^T\mathbf{x}) = \delta_i \mathbf{U} \left( \sum_{k=0}^{K} \eta_{i,k} T_k(\tilde{\Lambda}) \right) \mathbf{U}^T \mathbf{x} = \delta_i \sum_{k=0}^{K} \Psi_{ik} T_k(\tilde{\mathbf{L}})\mathbf{x} \tag{10}$$

where $\Psi = [\eta_{i,k}]_{ik} \in \mathbb{R}^{n \times (K+1)}$ is the trainable coefficient matrix, $\tilde{\mathbf{L}} = \mathbf{U}\tilde{\Lambda}\mathbf{U}^T$. Meanwhile, similar to the above discussion on localized filter, we have the following Proposition4.1 to claim that the adaptively filtered signal $\mathbf{z}$ is also approximately positioned around the node $i$.

**Proposition 4.1.** *Given a signal* $\mathbf{x}$ *defined on a graph* $\mathcal{G}$ *and a filter* $\mathbf{T}_i(\mathbf{g})$ *that translated to a given center node $v_i$ in* $\mathcal{G}$, *the filtered signal* $\mathbf{z} = \mathbf{x} * \mathbf{T}_i(\mathbf{g})$ *is approximately localized around the node $i$.*

To prove the proposition 4.1, let's first introduce the following Lemma 4.1.

**Lemma 4.1** ( [35]). *Let* $\hat{p}_K$ *be the polynomial approximation with degree K to the spectrum of a graph signal $\varphi$, i.e.,* $\hat{\varphi}(\lambda_l) \approx \hat{p}_K(\lambda_l) = \sum_{k=0}^{K} \gamma_k \lambda_l^k$. *If* $d_{\mathcal{G}}(i, n) > K$, *then* $\mathbf{T}_i(\varphi)_n \approx \mathbf{T}_i(p_K)_n = 0$, *where $p_K$ denotes the signal corresponding to $\hat{p}_K$.*

*Proof.* According to Definition 4.1 and the properties of convolution, we notice that $\mathbf{z}$ can be rewritten as:

$$\mathbf{z} = \mathbf{x} * \sqrt{N}(\mathbf{g} * \delta_i) = \sqrt{N}((\mathbf{x} * \mathbf{g}) * \delta_i) = \mathbf{T}_i(\mathbf{x} * \mathbf{g}) \tag{11}$$

Furthermore, let $\varphi = \mathbf{x} * \mathbf{g}$ and $\hat{p}_K$ be the polynomial approximation with degree K to $\hat{\varphi}$. From Lemma 4.1, $\mathbf{T}_i(\varphi)_n \approx \mathbf{T}_i(p_K)_n = 0$ will hold if $d_{\mathcal{G}}(i, n) > K$. Then, we have $\mathbf{z}_n = \mathbf{T}_i(\varphi)_n \approx 0$ if $d_{\mathcal{G}}(i, n) > K$.

This completes the proof. $\qquad\square$

---

**Algorithm 1** Node-oriented Spectral Filtering for GNNs

---
**Input:** $\mathbf{X} \in \mathbb{R}^{n \times f}, \tilde{\mathbf{L}} \in \mathbb{R}^{n \times n}, \text{K}$
**Output:** $\mathbf{Z}$
**Learnable Parameters:** $\mathbf{W}, \Gamma, \Theta$.
 1: $\mathbf{X}^{(0)} \leftarrow \text{MLP}_\Theta(\mathbf{X}), \quad \mathbf{X}^{(1)} \leftarrow \tilde{\mathbf{L}}\mathbf{X}. \quad$ /* *Feature Transformation* */
 2: **for** $k = 1$ to $K + 1$ **do**
        $\mathbf{X}^{(k)} \leftarrow 2\tilde{\mathbf{L}}\mathbf{X}^{(k-1)} - \mathbf{X}^{(k-2)}$ if $k > 1$
        **for** $i = 1$ to $n$ **do**
            $\eta_{i,k} \leftarrow \mathbf{H}_{i:}\Gamma_{:k} \quad$ /*$\mathbf{H} = \sigma(\mathbf{X}^{(k)}\mathbf{W})$*/
            $\mathbf{Z}_i \leftarrow \eta_{i,k}\mathbf{X}_i^{(k)} + \mathbf{Z}_i$
        **end for**
    **end for**

---

## 4.3 The Implementation of NFGNN

According to the proposed adaptive localized filtering in Eq 10, we will further formalize the architecture of the proposed NFGNN. As pointed out in [41, 20], the entanglement of feature transformation and filtering may be harmful to the performance and robustness of the GNN model. Hence, we adopt the similar way by first applying a MLP to perform the non-linear transformation for the raw feature matrix $\mathbf{X}$. Then, the spectral filtering operation can be implemented by a recursive way due to the stable recurrence relation of $T_k(\cdot)$:

$$T_k(\tilde{\mathbf{L}}) = 2\tilde{\mathbf{L}}T_{k-1}(\tilde{\mathbf{L}}) - T_{k-2}(\tilde{\mathbf{L}}) \tag{12}$$

Accordingly, given the input $X$, we will have $\mathbf{X}^{(k)} = T_k(\tilde{\mathbf{L}})\mathbf{X}$.

Notice that, the scale of the trainable coefficient matrix $\Psi$ is positive proportional to the number $n$ of nodes. With the increase of $n$, this will inevitably involve learning a large number of parameters. At the same time, the model also needs to learn $K$ parameters for a single node, which is intractable to be optimized. In addition, learning such a large number of parameters can also lead to overfitting, especially in the case of small number of labels. To achieve parameter lightweight for $\Psi$, we use a separable low-rank approximation to re-parameterize it. Specifically, $\Psi$ is assumed to be decomposed into two trainable parameter matrices $\Psi = \mathbf{H}\Gamma$ with $\eta_{i,k} = H_{i:}\Gamma_{:k}$, where $\mathbf{H} \in \mathbb{R}^{n \times d}$ and $\Gamma \in \mathbb{R}^{d \times (K+1)}$ are the node-dependent matrix and node-agnostic matrix, respectively.

As $\Gamma$ is seen as node-agnostic, it can be directly trained as general parameters, which is very similar to the learning of the polynomial coefficients in [7]. But for $\mathbf{H}$, since we treat it as node dependent, a simple yet effective nonlinear transformation (MLP) is applied, i.e., $\mathbf{H} = \sigma(\mathbf{X}\mathbf{W})$, where $\mathbf{W} \in \mathbb{R}^{f \times d}$ and $\sigma(\cdot)$ are the learnable weight matrix and activation function, respectively. It is worthy of note that, through low-rank approximation based re-parameterization, the parameter complexity of $\Psi$ is reduced from $\mathcal{O}(n \times (K+1))$ to $\mathcal{O}((K+1) \times d + f \times d)$, and we can flexibly adjust the model capacity by changing $d$. Particularly, we set $d = 1$ in this work, and thus the node-agnostic matrix $\Gamma \in \mathbb{R}^{1 \times (K+1)}$ is closely related to $\{\gamma_k\}_{k=0}^K$ in Eq. 6. We summarize the proposed node-oriented filtering in Algorithm1. [3]

## 5 Experimental Results and Analysis

### 5.1 Experimental Settings

**Datasets.** To provide a comprehensive evaluation of our method, several graphs from various domains with different homophily ratios are used, including 5 homophilic graphs: citation graphs Cora, CiteSeer, PubMed [31], and co-purchase graphs Computers and Photo [32]; 5 heterophilic graphs: Wikipedia graphs Chameleon and Squirrel [30], the Actor cooccurrence graph, and webpage graphs Texas and Cornell from WebKB [28]. The statistics of these datasets are summarized in supplemental materials.

**Baselines.** Several baselines have been selected for comparison, including 6 methods that can be seen as spectral filtering based, GCN [19], SGC [41], ChebNet [7], APPNP [20], GPRGNN [6],

---

[3]The source code for the implementation of NFGNN can be seen in supplemental materials.

Table 1: Results on real-world graphs: Mean accuracy (%) $\pm$ 95% confidence interval. Boldface letters mark the best result, while underlined letters denote the second best result.

| | Cora | Citeseer | PubMed | Computers | Photo | Chameleon | Actor | Squirrel | Texas | Cornell |
|---|---|---|---|---|---|---|---|---|---|---|
| **NFGNN** | 77.69±0.91 | 67.74±0.52 | **85.07±0.13** | **84.18±0.40** | **92.16±0.82** | **72.52±0.59** | **40.62±0.38** | 58.90±0.35 | **94.03±0.82** | 91.90±0.91 |
| BernNet | 76.37±0.36 | 65.83±0.61 | 82.57±0.17 | 79.57±0.28 | 91.60±0.35 | 68.73±0.57 | 40.01±0.42 | 50.75±0.67 | 92.30±1.23 | **91.96±1.07** |
| GPRGNN | **79.51±0.36** | 67.63±0.38 | **85.07±0.09** | 82.90±0.37 | 91.93±0.26 | 67.48±0.40 | 39.30±0.27 | 49.93±0.53 | 92.92±0.61 | 91.36±0.70 |
| APPNP | 79.41±0.38 | **68.59±0.30** | 85.02±0.09 | 81.99±0.26 | 91.11±0.26 | 51.91±0.56 | 38.86±0.24 | 34.77±0.34 | 91.18±0.70 | 91.80±0.63 |
| ChebNet | 71.39±0.51 | 65.67±0.38 | 83.83±0.12 | 82.41±0.28 | 90.09±0.28 | 59.96±0.51 | 38.02±0.23 | 40.67±0.31 | 86.08±0.96 | 85.33±1.04 |
| SGC | 70.81±0.67 | 58.98±0.47 | 82.09±0.11 | 76.27±0.36 | 83.80±0.46 | 63.02±0.43 | 29.39±0.20 | 43.14±0.28 | 55.18±1.17 | 47.80±1.50 |
| GCN | 75.21±0.38 | 67.30±0.35 | 84.27±0.01 | 82.52±0.32 | 90.54±0.21 | 60.96±0.78 | 30.59±0.23 | 45.66±0.39 | 75.16±0.96 | 66.72±1.37 |
| LINKX | 62.40±1.37 | 55.94±0.96 | 84.33±0.02 | 73.64±0.57 | 79.84±1.21 | 69.97±0.44 | 39.22±0.72 | 58.31±0.47 | 90.33±0.41 | 87.36±1.00 |
| BMGCN | 74.07±0.25 | 64.34±0.92 | 84.71±0.34 | NA | NA | 69.69±1.21 | NA | 53.16±0.74 | 93.00±0.57 | NA |
| FAGCN | 78.10±0.21 | 66.77±0.18 | 84.09±0.02 | 82.11±1.55 | 90.39±1.34 | 61.59±1.98 | 39.08±0.65 | 44.41±0.62 | 89.61±1.52 | 88.52±1.33 |
| GeomGCN | 20.37±1.13 | 20.30±0.90 | 58.20±1.23 | NA | NA | 61.06±0.49 | 31.81±0.24 | 38.28±0.27 | 58.56±1.77 | 55.59±1.59 |
| GAT | 76.70±0.42 | 67.20±0.46 | 83.28±0.12 | 81.95±0.38 | 90.09±0.27 | 63.9±0.46 | 35.98±0.23 | 42.72±0.33 | 78.87±0.86 | 76.00±1.01 |
| MLP | 50.34±0.48 | 52.88±0.51 | 80.57±0.12 | 70.48±0.28 | 78.69±0.30 | 46.72±0.46 | 38.58±0.25 | 31.28±0.27 | 92.26±0.71 | 91.36±0.70 |
| LINK | 42.94±2.02 | 25.52±1.98 | 54.78±0.96 | 70.05±1.31 | 78.84±1.45 | 71.09±1.16 | 26.25±1.43 | **59.77±1.27** | 89.61±1.52 | 44.91±2.19 |

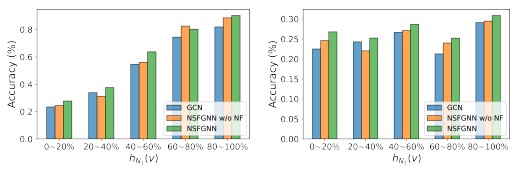 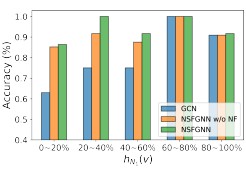 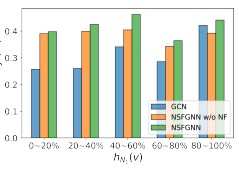

(a) Cora      (b) Citeseer      (c) Cornell      (d) Actor

Figure 3: Mean classification accuracy of nodes range by homophily ratio $h_{N_1}(v)$ on four datasets.

BernNet [13], and 3 spatial aggregation based methods, GAT [38], Geom-GCN [28], BMGCN [10], and 3 non-GNN baselines, MLP, LINK [43], and LINKX [22]. For GPRGNN, BernNet, and BMGCN, we directly use the open-source codes released by the original paper. For the others, we use the models that are provided by [6].

**Experimental Setup.** For the node classification task, we follow the random split ratio in [6] to split the dataset into training/validation/test sets. Specifically, The sparse splitting ratio (2.5%/2.5%95/%) is used for homophilic graphs, and the dense splitting ratio (60%/20%/20%) is used for heterophilic graphs. We run each experiment 50 times with random initialization, and random data splits. Finally, we report the average results with a 95% confidence interval. We set the degree of the polynomial $K = 10$ for all datasets. The Adam [18] is employed as the optimizer for the NFGNN training. For GPRGNN, BernNet, and BMGCN, we use the best combination of hyperparameters provided in the original paper to report the results for each dataset.[4]

## 5.2 Performance Comparison

The average results of running 50 times on the node classification task are reported in Table 1, where accuracy is used as the evaluation metric with a 95% confidence interval. NFGNN outperforms all the baselines on 6 datasets and achieves comparable results on the other 4 datasets. In particular, on Chameleon and Squirrel graphs, NFGNN outperforms the SOTA method BMGCN by a large margin, i.e., 2.83% and 5.03%, demonstrating the superiority of our method.

Meanwhile, it can be observed several interesting phenomena in Table 1. **i**) GCN and GAT are even inferior to MLP on some heterophilic graphs, which shows that positive near-neighbor aggregation is indeed out of power in some cases. Besides, the performance of MLP also shows that the utilization of node features is also very important for GNNs. **ii**) The filter-learning based methods generally have good a performance on both the homophilic and heterophilic graphs, indicating that adaptive filter learning has better transferability than filter pre-designing.

## 5.3 Node-level Analysis

A motivation of the proposed NFGNN is to solve the mixed local patterns discussed in Sect. 3. Therefore, we divide the test nodes into 5 different intervals according to the homophilic 1-hop neighbor ratio $h_{N_1}(v)$ and report the mean accuracy of each interval. The results of GCN, NFGNN

---

[4]More detailed experimental settings are discussed in the supplementary material.

Table 2: Accuracy (%) improvement of the node-oriented filtering (NF).

| Basis | | Cora | Citeseer | Pubmed | Chameleon | Actor | Texas |
|-------|--------|--------|----------|--------|-----------|--------|--------|
| Monomial | w/o NF | 78.15 | 66.60 | 82.28 | 61.79 | 38.88 | 91.80 |
| | w/ NF | 79.16 | 68.42 | 84.79 | 63.36 | 39.53 | 91.47 |
| | Improv. | (1.01) | (1.82) | (2.51) | (1.57) | (0.66) | (-0.33) |
| Bernstein | w/o NF | 76.32 | 65.61 | 82.10 | 67.82 | 39.31 | 92.29 |
| | w/ NF | 78.41 | 66.22 | 83.09 | 69.93 | 40.77 | 93.37 |
| | Improv. | (2.09) | (0.61) | (0.99) | (2.11) | (1.46) | (1.08) |
| Chebyshev | w/o NF | 76.07 | 65.11 | 84.02 | 68.48 | 39.11 | 92.47 |
| | w/ NF | 77.69 | 67.74 | 85.07 | 72.52 | 40.62 | 94.03 |
| | Improv. | (0.62) | (2.63) | (1.05) | (4.04) | (1.51) | (1.56) |

with only $\gamma$ (marked as NFGNN w/o NF) and NFGNN are shown in Fig 3. It should be noticed that the NFGNN w/o NF is equivalent to learning a globally consistent filter using the Chebyshev polynomial. It can be seen from Fig. 3(c) and (d) that NFGNN has a promising and similar performance on all five intervals, which shows that NFGNN can effectively capture the various local patterns under the condition as long as the amount of trainable data is sufficient. Besides, both NFGNN and NFGNN w/o NF perform better than GCN on the semi-supervised node classification task, as shown in Fig. 3(a) and (b). It suggests that adaptive learning filters are no less expressive than pre-designed filters, even in the semi-supervised case.

## 5.4 Effectiveness of the Node-oriented Filtering

To evaluate the effectiveness of the proposed node-oriented filtering more comprehensively, we first compare the performance of NFGNN and NFGNN w/o NF. Further, since the node-oriented filtering is independent of the polynomial basis, the Chebyshev basis is replaced by the Monomial basis and Bernstein basis, respectively, and we check the improvement bring by the node-oriented filtering mechanism for them. For the Bernstein basis, we refer to the implementation form given in [13]. The results on six graphs are summarized in 5.4. Firstly, it can be seen that the globally consistent filters learned using three different bases have leading performance on different datasets, respectively, illustrating the effectiveness of using a polynomial approximation to learn filters. Furthermore, except for the Monomial basis on Texas graph, the node-oriented filtering mechanism has different enhancements for each basis. The improvements not only validate the effectiveness of the proposed node-oriented filtering, but also demonstrate that the polynomial filter and the node-oriented filtering can each other to some extent.

## 6 Conclusion and Discussion

In this paper, we first analyze in depth the local patterns in graph data and the aggregatability of Near-neighbors. Motivated by these observations, we rethink the spectral-based GNNs and propose NFGNN for node-oriented spectral filtering via the generalized translated operator. Compared to previous methods that learn a global filter, NFGNN performs spectral filtering through filters translated on specific nodes to address the issue of local patterns. Through recursive form and re-parameterization trick, the oriented-filtering is implemented in a simple way. The experimental results on several real-world graph datasets verify that our NFGNN achieves more remarkable performance over currently available alternatives.

Studying spectral-based GNNs in accordance with the idea of graph signal processing theory is one of the origins of GNNs. With different starting points, the spatial-designed GNNs aim to design the neighborhood aggregation mechanism based on the topological characteristics of the graph, focusing more on the local relationship between nodes and their neighbors. In contrast, spectral-based GNNs are dedicated to the design of the filtering of the graph signal in the spectral domain, analyzing the graph more from a global perspective. The proposed NFGNN in this paper provides a new form of trade-off between global and local perspectives in the spectral domain. Particularly, NFGNN can be seen as a extension of the existing methods for estimating global filters. For spectral-based GNNs, the scalability of spectral convolution and inductive learning setting are still key issues to be solved at present, and it is still one of the directions of graph neural networks that can be expected because of the great transferability it exhibits.

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
