# OpenReview forum: "Node-oriented Spectral Filtering for Graph Neural Networks"
_NeurIPS.cc/2022/Conference — NeurIPS 2022 Submitted_

### Official Review · Reviewer_WEKa · 2022-07-08

**Rating:** 4
**Confidence:** 3
**Soundness:** 2 fair
**Presentation:** 2 fair
**Contribution:** 2 fair

**Summary:**

This paper discusses node oriented spectral filtering as an extension to Chebyshev polynomial of normalized adjacency matrix. As motivation the paper argues on the aggregation of neighbors of labeled nodes. The paper argues on superiority on non-homophilic graphs through experiments.

**Questions:**

The argument of 2-hops and 3-hops aggregation is a bit confusing. It is mainly based on experiments using a few set of datasets. To my knowledge, methods such as the GPRGNN and Mixhop which uses higher-order convolutions (acting as convolution from multiple hops) do not impose such limits. Can authors elaborate on this?


This paper only contain experiments with graph data by [Pei et al]. Further, the homophily measure described in the paper is a bit dated.
There is a new homophily measure and new non-homophilic benchmark datasets introduced by [Lim et al., 2021a, Lim et al., 2021b].
It would be interesting know how the proposed method perform with some of the new non-homophilic data. Can the authors provide few more experiment with the new dataset?

Two important non-GCN/GNN bechmark method that are important for comparison are the MLP and LINK [Zheleva and Getoor, 2009]. MLP would allow us to know about learning ability of node features itself and LINK tells us about direct learning from graph data only (adjacency matrix). Can the authors add these two benchmark methods in the paper? Another benchmark method that is useful for comparison is LINKX [Lim et al., 2021a].


Reference

[Zheleva and Getoor, 2009] Zheleva, E. and Getoor, L. (2009). To join or not to join: The illusion of privacy in  social networks with mixed public and private user profiles. WWW ’09.

 [Lim et al., 2021a] Lim, D., Hohne, F. M., Li, X., Huang, S. L., Gupta, V., Bhalerao, O. P., and Lim, S.-N.  (2021a). Large scale learning on non-homophilous graphs: New benchmarks and strong simple methods. In  NeurIPS.

 [Lim et al., 2021b] Lim, D., Li, X., Hohne, F., and Lim, S.-N. (2021b). New benchmarks for learning on  non-homophilous graphs. Workshop on Graph Learning Benchmarks, WWW 2021.

**Limitations:**

Limitations are not well discussed.

**Strengths And Weaknesses:**

Strengths:
1.) Experimental results are promising

Weaknesses:
1.) Highly Incremental work. Very closely related to ChebNet and related method.
2.) Homophily definition is not up to date and experiments are not adequate.

---

> ### Author Response · Authors · 2022-08-02
> **Author Response to Reviewer WEKa: Table R4-1, Table R4-2**
>
>   Table R4-1. Comparison to MLP, LINK, and LINKX under the experimental setup of this paper: Mean accuracy (%) ± 95% confidence interval.
> |Methods|Cora|Cite.|Pub.|Comp.|Photo|Cham.|Actor|Squi.|Texas|Corn.|
> |:-|:-:|:-:|:-:|:-:|:-:|:-:|:-:|:-:|:-:|:-:|
> |MLP|50.34±0.48|52.88±0.51|80.57±0.12|70.48±0.28|78.69±0.30|46.72±0.46|38.58±0.25|31.28±0.27|92.26±0.71|91.36±0.70|
> |LINK  |42.94±2.02|25.52±1.98|54.78±0.96|70.05±1.31|78.84±1.45|71.09±1.16|26.25±1.43|**59.77±1.27**|89.61±1.52|44.91±4.19|
> |LINKX  |62.40±1.37|55.94±0.96|82.33±0.02|73.64±0.57|79.84±1.21|69.97±0.44|39.22±0.72|58.31±0.47|90.33±0.41|87.36±1.00|
> |NFGNN|**77.69±0.91**|**67.74±0.52**|**85.07±0.40**|**84.18±0.40**|**92.16±0.82**|**72.52±0.59**|**40.62±0.38**|58.90±0.35|**94.03±0.82**|**91.90±0.91**|
>
>
> Table R4-2. Results on the datasets proposed by [Lim et al., 2021a]
> |Models|Penn94|pokec|arXiv-year|snap-patents|genius|twitch-gamers|
> |:-|:-:|:-:|:-:|:-:|:-:|:-:|
> |LINKX|84.71±0.52|82.04±0.07|56.00±1.34|61.95±0.12|90.77±0.27|66.06±0.19|
> |GPRGNN|81.38±0.16|78.83±0.05|45.07±0.21|40.19±0.03|90.05±0.31|61.89±0.29|
> |NFGNN|84.10±0.40|81.37±0.11|49.95±0.32|52.23±0.22|90.94±0.43|64.13±0.16|

---

> > ### Comment · Reviewer_WEKa · 2022-08-09
> > **Reply**
> >
> > Thank you for the reply. However, I am a bit concerned with the results from the proposed method on [Lim et al. 2021b]. Many existing methods that uses work well for non-homophilous data by Pei et al. and extensions such as NFGNN naturally work well on these datasets. The challenge is to develop model that can do well with data such as [Lim et al. 2021b].

---

> > > ### Author Response · Authors · 2022-08-09
> > > **Author Response**
> > >
> > > We totally agree that the datasets proposed in [Lim et al. 2021a, 2021b] are more challenging compared to the existing heterophily graph datasets. As we mentioned in the previous reply, the new benchmark datasets are an excellent complement to the traditionally used ones, which can further promote the research of heterophily graphs in the academic community.
> > >
> > > As presented in Table R4-2, for the results on the new datasets under the supervised learning setting as [Lim et al. 2021a], the performance of NFGNN is indeed weaker than LINKX on arxiv-year, snap-patents, and twitch-gamers datasets. However, we also noticed that our NFGNN is comparable to the strong baseline LINKX on the 3 datasets, penn94, pokec, and genius datasets. In addition, NFGNN also shows a significant improvement over GPRGNN, which is also one of the SOTA spectral-based GNNs.
> > >
> > > Besides, compared with the datasets proposed by Pei et al., one of the characteristics of the new datasets [Lim et al. 2021a,b] is a much larger scale. Generally, there are only a few labels are available for such large-scale graphs in real application scenarios. Therefore, we also conduct experiments on semi-supervised learning on these datasets [Lim et al. 2021a,b]. Under the semi-supervised learning setting with sparse splitting (2.5% for training, 2.5% for validation, 95% for testing), the performance of NFGNN and LINKX are reported in Table R4-3. Specifically, the optimal hyperparameters of LINKX are obtained using the same way as the ones in fully supervised learning, and the hyperparameters of NFGNN are the same as the settings for fully supervised learning without further tuning due to the time limitation.
> > > It can be seen from Table R4-3 that the gap between the two approaches is further narrowed compared to Table R4-2, and even NFGNN surpasses LINKX on 4 of 6 datasets. Particularly, on the snap-patents dataset, NFGNN is worse than LINKX under the fully supervised learning setting, while it outperforms LINKX under the semi-supervised learning setting. Overall, the performance of NFGNN degrades less than LINKX with fewer labels. In practice, if we can spend more time tuning the hyperparameters, better performance can be expected for our NFGNN.
> > >
> > > In the case of generality, just as you mentioned, it is really a challenge to develop models that can perform well on new datasets. As shown in Table R4-1, R4-2, and R4-3, GPRGNN achieves strong performance on the datasets proposed by Pei et al., but it fails to perform as well on the new datasets [Lim et al.].
> > > Contrary to GPRGNN, LINKX performs well on the new datasets [Lim et al.] but is not satisfactory on the datasets proposed by Pei et al. Unlike them, NFGNN not only achieves excellent performance on the datasets [Pei et al.], but also is comparable to LINKX and significantly better than GPRGNN on the new datasets [Lim et al.]. Based on the above observation, it shows that the proposed NFGNN indeed has good generality to some extent. Besides, we also note that LINKX has explicitly utilized the adjacency matrix as additional features and achieved good performance on the new datasets without further using aggregation operations like the conventional GNNS. Inspired by it, it may be a good attempt to incorporate this highlight into some other aggregation-based GNN models including our NFGNN to further improve performance on heterophily graphs.
> > >
> > > **TableR4-3**:
> > > Results on the datasets proposed by [Lim et al., 2021a] under the semi-supervised learning setting with sparse dataset splitting (2.5% for training, 2.5% for validation, 95% for testing).
> > > |Models|Penn94|pokec|arXiv-year|snap-patents|genius|twitch-gamers|
> > > |:-|:-:|:-:|:-:|:-:|:-:|:-:|
> > > |LINKX|64.85±0.79|64.99±2.36|41.04±0.55|40.38±1.12|88.15±0.16|63.83±0.44|
> > > |NFGNN|66.50±2.14|74.86±0.16|37.77±0.40|43.54±0.32|90.41±0.19|62.91±0.79|

---

> > > > ### Comment · Reviewer_WEKa · 2022-08-10
> > > > **Reply**
> > > >
> > > > Thank you for the update. Though the sparse splitting gives a descent performance, I am still not very impressed since homophily is a challenging problem and new methods should be strong enough to handle them.

---

> > > > > ### Author Response · Authors · 2022-08-10
> > > > > **Reply to Reviewer WEKa**
> > > > >
> > > > >
> > > > >   We sincerely thank the reviewers for their comments and regret not having completely eliminated your concerns about the ability of our NFGNN model in handling heterogeneous graphs. In the new comment, the reviewer thinks that, for a challenging complex graph mixing of homophily and heterophily, the new methods should be strong enough to handle them.
> > > > > We partly agree with the reviewer on this view for evaluating work, and in practice, our model also archives good performance and even significant improvement compared with the popularly adopted baselines, which have been well presented in the original manuscript and our replies to the previous comments.
> > > > > However, we are not quite convinced that the substantial contributions of a work submitted to an academic conference, e.g., NeurIPS, should be evaluated simply and solely from a performance improvement perspective. In fact, in addition to the concerns about performance improvement, more insightful comments from the aspects of novelty and theoretical or technical contributions may be more constructive and appreciated.
> > > > >
> > > > > As emphasized in the previous responses, our proposal is not only concerned with the performance of the proposed model on the heterophily graphs, but also opens up a promising new way for spectral-based GNNs to learn local filters. Besides, the proposed NFGNN not only maintains the existing good performance on homophily graphs, but also makes considerable progress on heterophily graphs compared with existing spectral-based GNNs. Meanwhile, the theoretical and empirical analysis of the homophily and heterophily of the datasets presented in this paper is also one of the contributions that can’t be ignored. Therefore, we argue that this paper provides a noteworthy contribution to the academic community in addition to performance improvements.
> > > > >
> > > > > In addition, even as a new strong baseline model, the LINKX doesn’t show a universal excellent performance as expected, e.g., on the datasets proposed by Pei et al. Under a more realistic semi-supervised learning, our NFGNN still outperforms LINKX overall as we can see from Table R4-3. As we know, at least for now, it is objectively difficult for any one model to perform superbly in all situations. To promote progress in dealing with complex graph data, it should be encouraged to explore new ideas and means both in theory and practice ( or performance ) to solve the challenges in graph learning, which is also the goal of this paper.

---

> ### Author Response · Authors · 2022-08-02
> **Author Response to Reviewer WEKa**
>
> We thank the reviewer for the constructive comments. The responses are listed below:
>
>  **Q1: Highly Incremental work. Very closely related to ChebNet and related method.**
>
> A1: As one of the spectral-based methods, we agree that there inevitably exist some close connections of the proposed NFGNN model with other spectral-based GNNs including ChebNet. However, as pointed out by reviewer 3, we believe that the proposed node-oriented filtering paves a new perspective for local filter learning with good theoretical properties. It also allows the spectral-based GNNs to no longer be limited to learning globally shared filters. To some extent, the proposed work can’t be simply seen as an incremental yet effective work. Essentially, it can help in improving the generalizability of the family of spectral GNNs, which is very necessary for many real practical situations.
>
> **Q2: Homophily definition is not up to date.**
>
> A2: Thanks for your valuable suggestions. In this paper, we mainly follow the widely adopted homophily measure defined in [Pei et al., ICLR2019]. We are sorry for missing the new work by [Lim et al. 2021b]. After reading it, we totally agree that the new homophily measure has better properties that are less sensitive to the number of classes and size of each class than the one by Pei et al. To further improve the rigor and comprehensiveness of our paper, we have included a discussion of this measure in the revised version.
>
> **Q3: The argument of 2-hops and 3-hops aggregation is a bit confusing. Methods such as the GPRGNN and Mixhop which uses higher-order convolutions do not impose such limits. Can authors elaborate on this?**
>
> A3: We think what we stated is not in conflict with your comments. Here, both the 2-hops and 3-hops aggregation refer to the aggregation of classical GNNs, which employ averaging or positive weighted averaging for neighbor aggregation, such as GCN, GAT, and GraphSAGE. As well known, when simply stacking multiple propagation layers, this type of aggregation has been shown to be prone to the serious over-smoothing problem. Therefore, such GNNs are generally designed as shallow networks and thus lack the ability to capture long-distance neighborhood information. In addition, combined with our analysis of the homophily of neighborhoods as given in Sect.3.2, it can be known that these GNN models also suffer from difficulty in capturing enough information in non-homophilous graphs.
>
> As you mentioned, GPRGNN and MixHop were also proposed to address this problem faced by the classical GNNs from different perspectives. In fact, GPRGNN allows learning GPR weights to aggregate representations between different layers, and MixHop adopts concatenation to mix the multiple powers of the adjacency matrix. Both of them are not limited to near-neighbor aggregation even up to 3 or more hops, and thus can handle better non-homophilous graphs than the classical GNNs.
>
> **Q4: Can the authors add these two benchmark methods and LINKX [Lim et al., 2021a] in the paper?**
>
> A4: Thanks for the insightful suggestions. In essence, both the LINK and the LINKX proposed recently can be seen as non-GCN-like benchmark methods and deserve to make the comparison as suggested. We added the performances of LINK and LINKX on both homophily and non-homophily datasets in Table 1 (i.e., the following Table R4-2) of the revised version. Besides, the results of MLP has ever been reported in Table 1 of the original version. As we can see, as a simple baseline, LINK achieves the best performance on the Squirrel dataset. Besides, all three baselines being compared perform poorly on homophily graphs under semi-supervised learning settings.
>
> **Q5: Can the authors provide few more experiment with the new datasets by [Lim et al., 2021a]?**
>
> A5: According to your suggestion, we also conduct additional experiments on the new large-scale non-homophilic datasets proposed by [Lim et al. 2021a] to further verify the generalization ability of our NFGNN. We strictly follow the experimental setup in [Lim et al., 2021a] and present the results in Table R4-2. Although our method achieves slightly inferior performance compared to LINKX, it has still a significant improvement over GPRGNN. Meanwhile, the results of Tables R4-1 and R4-2 also illustrate that, unlike on the traditionally used datasets in Table R4-1, LINKX achieves the best performance among the compared baselines on the newly released datasets. It means the new benchmark datasets are also a good complement to the traditionally used ones.
>
> **Q6: Limitations are not well discussed.**
>
> A6: For the proposed NFGNN, the scalability of spectral convolution and the inductive learning setting are still the same key issues as faced in other spectral-based GNNs. Considering the transferability in real applications, some further studies are also needed to address these issues. We added the discussions on the above limitations in Sect.6 of the revised version.

---

### Official Review · Reviewer_TWyA · 2022-07-11

**Rating:** 7
**Confidence:** 3
**Soundness:** 4 excellent
**Presentation:** 4 excellent
**Contribution:** 2 fair

**Summary:**

In this work the authors address the problem of local structure variation and non-homophily by introducing a generalized translation operator in order to implement a node centered spectral filter for graph neural networks. They demonstrate the empirical effectiveness of their method by confirming the realization of the filter localization in practice and showing performance gains on datasets with different homophily ratios.

**Questions:**

1. I'm a little confused by the wording of line 224 - it's stated as if $\Psi$ should be included in Eq. 9 somewhere?
2. It is somewhat out of scope I'm sure, but as a convincing sanity check, it would be nice to have the _complement_ to the demonstration that the NF method indeed improves node oriented predictions, the complement being how it effects the full graph classification task. Would your hypothesis be that it doesn't improve/alter global representations much in that setting? (this would be a meta ablation of sorts)

**Limitations:**

1. Barely a limitation, more a comment, that Table 1 is not as interesting/relevant to me as Fig 3 and Table 2. The direct analysis on the ability of the NF method to address heterophily should be the focus of the empirical section, even extended if possible.


**Strengths And Weaknesses:**

### Strengths

**Quality/Clarity**: The preliminaries and technical details of their proposed method are sound. The motivations section is particularly in depth as it provides not only theoretical but also empirical analysis of the homophily versus heterophily characteristics of the datasets to be used in the evaluation.

**Significance**: Though the modification is small, it fundamentally increases the adaptiveness of the family of spectral GNN approaches that, while having some nice theoretical properties, in their standard formulation, lack fine grained expressiveness or the ability to scale to large scale graphs where global filters are likely not as meaningful. Figure 3 and Table 2 provide targeted evidence of the effectiveness of the translation operator/NF modification.

### Weaknesses

**Significance**: The margins of improvement in Table 1 are not very significant except in Squirrel and Texas

**Clarity**: There is not adequate discussion devoted prior to Section 5 in the preliminaries or the other theoretical prose on how exactly the localized filter is expected to better address variations in homophily characteristics. This is somewhat borne out by the results anyway, but it should be motivated earlier on.

**Originality**: Also relating to the significance, the fact that the core modification itself is quite compact, is noted, and thus the likelihood that the approach is completely novel is potentially low. But this isn't a focus of this review.

---

> ### Author Response · Authors · 2022-08-02
> **Author Response to Reviewer TWyA**
>
> We thank the reviewer for the constructive comment and positive recognition of our work. The responses are listed below:
>
> **Q1: The margins of improvement in Table 1 are not very significant except in Squirrel and Texas.**
>
> A1: We strongly agree that, although our NFGNN achieves better performance compared to the existing baselines, the performance gains in Table 1 are not adequately significant as expected. In fact, for the 10 popularly referred benchmark datasets in GNN evaluation, we have witnessed many SOTA works that have been verified on them, which makes large performance improvement on them very challenging. Even so, in addition to Squirrel and Texas datasets, there are also more 1% improvements over the best baselines on both the Computers and Chameleon datasets, showing that our method is somewhat competitive.
>
> **Q2: How exactly the localized filter can better address the variation of homophily characteristics is confirmed to some extent by the results, but should be motivated in front.**
>
> A2: According to your suggestion, we first made an explanation more intuitively in Sect.4 about why localized filters can better address the variation of homophily characteristics compared to global filter, and then the experimental results in Sect.5 are used to further confirm the effectiveness of the localized filter. In this way, the motivation for the node-oriented localized filter can be further strengthened.
>
> **Q3: I'm a little confused by the wording of line 224 - it's stated as if Ψ should be included in Eq. 9 somewhere?**
>
> A3: Thanks for pointing the writing mistake out. Eq.(9) has been modified properly.
>
> **Q4: How it effects the full graph classification task. Would your hypothesis be that it doesn't improve/alter global representations much in that setting?**
>
> A4: The same valuable suggestion was also pointed out by Reviewer 1. As we know, most of GNNs based on spectral graph theory were generally developed for the node classification task due to its intrinsic good interpretation ability. Unlike the node classification task, more works have been studied from the perspective of graph isomorphism for full graph classification. To the best of our knowledge, fewer works have also been focused on the connections between graph spectra and graph isomorphism [1,2,3]. Although there is no straightforward explanation, our intuition is that the expressive node representations would implicitly facilitate the learning of the full graph representation.
>
> According to your kind suggestion, the widely used TU benchmark containing 5 datasets for graph classification is adopted to validate the performance of NFGNN on graph classification tasks. To ensure the fairness of the experiment, we strictly follow the experimental setup in [4] and also used the baselines therein for comparison, which include GCN, GraphSAGE, GIN, and GAT. Besides, we also choose extra GPRGNN and BernNet for the graph classification task. As shown in Table R3-1, our NFGNN is also competitive on the graph classification task.
>
> Table R3-1. Results on TU datasets: Mean accuracy (%) ± standard deviation.
> |Methods|D&D|MUTAG|PROTEINS|PTC_MR|ENZYMES|
> |:-|:-:|:-:|:-:|:-:|:-:|
> |GCN|71.6±2.8|73.4±10.8|71.7±4.7|56.4±7.1|27.3±5.5|
> |GraphSAGE|71.6±3.0|74.0±8.8|71.2±5.2|57.0±5.5|30.7±6.3|
> |GIN|70.5±3.9|**84.5**±8.9|70.6±4.3|51.2±9.2|**38.3**±6.4|
> |GAT|71.0±4.4|73.9±10.7|72.0±3.3|57.0±7.3|30.2±4.2|
> |GPRGNN|75.3±5.6|75.0±8.0|72.1±3.3|60.3±4.0|27.8±5.6|
> |BernNet|74.7±4.4|74.5±7.4|72.2±2.6|58.3±7.6|27.2±5.4|
> |NFGNN|**75.9±5.5**|75.6±8.1|**72.4**±3.6|**62.5**±9.9|28.3±7.1|
>
> [1] Edwin et al. Which graphs are determined by their spectrum? Linear Algebra and its applications, 373:241–272, 2003.
>
> [2] Rattan et al. Weisfeiler--Leman, Graph Spectra, and Random Walks. arXiv:2103.02972, 2021.
>
> [3] Fiori et al. On spectral properties for graph matching and graph isomorphism problems. Information and Inference: A Journal of the IMA. 2015.
>
> [4] Zhang et al. Nested Graph Neural Networks. NeurIPS2021.
>
> **Q7: Barely a limitation, more a comment, that Table 1 is not as interesting/relevant to me as Fig 3 and Table 2. The direct analysis on the ability of the NF method to address heterophily should be the focus of the empirical section, even extended if possible.**
>
> A7: Thanks for your insightful comments. Although we have endeavored to give a direct analysis of the ability of the NF method to address heterophily through Fig 3 and Table 2, your comment yet inspires us to make a further rethinking about it from some other aspects. For example, through the observation of the localized parameter matrix of the polynomial $\Psi$, it can be tried to find what kind of local patterns NFGNN has ever been learned. After the rebuttal period, we will make some empirical studies on it as soon as possible.

---

> ### Comment · Reviewer_TWyA · 2022-08-09
> **Acknowledgment of author rebuttal**
>
> Thanks to the authors for their detailed responses to my review.
>
> In particular, the additional experiments on graph classification are appreciated. The results are compelling - it would be interesting to analyze the potential intrinsic advantage GIN might have on those two datasets.
>
> Further analysis on the local structures represented after training would be very interesting! I will be on the look out for further work on the benefits of localized spectral approaches.

---

### Official Review · Reviewer_YZ7W · 2022-07-19

**Rating:** 7
**Confidence:** 3
**Soundness:** 3 good
**Presentation:** 3 good
**Contribution:** 3 good

**Summary:**

This work proposes a new graph neural network by designing a node-oriented spectral filtering method. The developed algorithm is motivated by the observation that the subgraphs centered at different nodes have different properties. Experimental results demonstrate superior performance compared to baseline models.

**Questions:**

See weakness section

**Limitations:**

see weakness section

**Strengths And Weaknesses:**

Strength:

1. This paper indicates the fact that real-world graphs are often a mixture of diverse subgraph patterns.
2. This paper shows that the standard near-neighborhood aggregation mechanisms fail to work for the non-homophily graph.
3. This paper proposes a novel node-oriented graph spectral filtering method, which gives rise to a more powerful GNN model.

Weakness:

Overall, I am satisfied with the experimental findings and observations. The proposed model seems to be novel and useful. Some minor issues are as follows:

1. What if using 3 or more hop neighborhood aggregation methods? Standard multi-layer GNN models implicitly perform multi-hop neighbor aggregation. A discussion on this point is needed.

2. Will using learnable node-oriented graph spectral filtering involve more trainable parameters? Will this incur more severe overfitting?

3. It would be better to give a more clear comparison between the proposed filtering method and the prior methods in Section 4.1 (i.e., give more equations) to better highlight the difference and novelty of the proposed method.

---

> ### Author Response · Authors · 2022-08-02
> **Author Response to Reviewer YZ7W.**
>
> We thank the reviewer for the constructive comment and positive recognition of our work. The responses are listed below:
>
> **Q1: What if using 3 or more hop neighborhood aggregation methods? Standard multi-layer GNN models implicitly perform multi-hop neighbor aggregation. A discussion on this point is needed.**
>
> A1: Thanks for your insightful suggestion. As you mentioned, standard multi-layer GNN models can perform multi-hop neighbor aggregation by stacking multiple layers. However,
> the averaging or positive weighted averaging aggregation employed by classical GNNs has been shown to seriously cause the expressive power of the model to degrade, i.e., over-smoothing, as the number of layers increases [1]. Therefore, classical GNNs, such as GCN, usually adopt shallow network architecture, which makes it difficult to effectively utilize information from distant neighborhoods. Meanwhile, as we discussed in Sect.3.2, the reason for the poor performance of classical GNNs on non-homophily graphs might be due to the insufficient homophilic information captured from the heterophily-preferred near-neighbors. Hence, we think that implementing effective long-range neighborhood aggregation, so as to fully utilize the information from long-distance nodes, will greatly benefit improving the performance of GNNs on non-homophily graphs.
>
> [1]Oono et al. Graph Neural Networks Exponentially Lose Expressive Power for Node Classification. ICLR2020.
>
> **Q2: Will using learnable node-oriented graph spectral filtering involve more trainable parameters? Will this incur more severe overfitting?**
>
> A2: Yes, as you mentioned, it will inevitably involve some more trainable parameters if we directly train them. But fortunately, due to the utilization of reparameterization of localized parameter matrix via the proposed low-rank decomposition into the node-dependent and node-agnostic matrices, it will only increase the parameters by a very small and controllable amount. In our case, since the rank-one approximation is used, i.e., d=1, the increased number of parameters with $\Psi$ will only be K+1+f in practice, which can be completely negligible compared to the total amount of model parameters. Overall, it means no more severe overfitting will be incurred.
>
> Meanwhile, compared to GPRGNN, our NFGNN model only brings f more parameters, and the total number of parameters can be much smaller than ChebNet. Hence, our NFGNN is also no worse than the existing spectral-based GNNs in terms of the issue of overfitting.
>
> **Q3: It would be better to give a more clear comparison between the proposed filtering method and the prior methods in Section 4.1 (i.e., give more equations) to better highlight the difference and novelty of the proposed method.**
>
> A3: Thanks for your valuable comments. In the supplement material, we have analyzed the connection and difference between several existing GNN models and ours, and summarized their polynomial filtering forms. Following your suggestion, we will try to introduce this part into the formal paper if space permits. Meanwhile, we first added a concise formula to briefly explain our NFGNN in Section 4.1 of the revised version, so as to better highlight the motivation and novelty.

---

### Official Review · Reviewer_LUHK · 2022-07-21

**Rating:** 4
**Confidence:** 4
**Soundness:** 3 good
**Presentation:** 3 good
**Contribution:** 2 fair

**Summary:**

The paper proposes a method called NFGNN which utilizes generalized translation operator to learn local patterns contributing to homophily/heterophily.


**Questions:**

Please see the section above.

**Strengths And Weaknesses:**

Strengths:
1. A new GNN method with ability to capture the local patterns.
2. Well organized and easy to follow (in general).


Weaknesses/Questions:

1. My main concern is the performance of the proposed method as compared to existing baselines.
Also, FAGCN adresses heterophily by considering low as well as high frequency information.
The numbers reported in FAGCN are superior to the proposed method.
Therefore it is difficult to establish the usefulness of the proposed method.

2. The evaluation on only node classification task seems limited to me.
Additional experiments on graph classification tasks will further strengthen the paper.

3. Even though Section 6 is well motivated, the content does not justify it to be a separate section.

4. Citation method is not proper. (please use \citep{} or \citet{} accordingly)

5. The purpose of reparameterization with low-rank approximation against first (or higher) order polynomial is not clear to me.

6. How does generalized translated operator help in adapting to local patterns? It is defined through spectral domain.
What is the intuition of this translation operator in spatial domain when a filter is translated to a node?

---

> ### Author Response · Authors · 2022-08-02
> **Author Response to Reviewer LUHK: Table R1-1, Table R1-2**
>
> Table R1-1: Comparison to FAGCN under the experimental setup of this paper: Mean accuracy (%) ± 95% confidence interval.
> |Methods|Cora|Cite.|Pub.|Comp.|Photo|Cham.|Actor|Squi.|Texas|Corn.|
> |:-     | :-:|:-:|:-:|:-:|:-:|:-:|:-:|:-:|:-:|:-:|
> |FAGCN  |**78.10±0.21**|66.77±0.18|84.09±0.02|82.11±1.55|90.39±1.34|61.59±1.98|39.08±0.65|44.41±0.62|89.61±1.52|88.52±1.33|
> |NFGNN|77.69±0.91|**67.74±0.52**|**85.07±0.40**|**84.18±0.40**|**92.16±0.82**|**72.52±0.59**|**40.62±0.38**|**58.90±0.35**|**94.03±0.82**|**91.90±0.91**|
>
> Table R1-2: Results on TU datasets: Mean accuracy (%) ± standard deviation.
> |Methods|D&D|MUTAG|PROTEINS|PTC_MR|ENZYMES|
> |:-|:-:|:-:|:-:|:-:|:-:|
> |GCN|71.6±2.8|73.4±10.8|71.7±4.7|56.4±7.1|27.3±5.5|
> |GraphSAGE|71.6±3.0|74.0±8.8|71.2±5.2|57.0±5.5|30.7±6.3|
> |GIN|70.5±3.9|**84.5**±8.9|70.6±4.3|51.2±9.2|**38.3**±6.4|
> |GAT|71.0±4.4|73.9±10.7|72.0±3.3|57.0±7.3|30.2±4.2|
> |GPRGNN|75.3±5.6|75.0±8.0|72.1±3.3|60.3±4.0|27.8±5.6|
> |BernNet|74.7±4.4|74.5±7.4|72.2±2.6|58.3±7.6|27.2±5.4|
> |NFGNN|**75.9±5.5**|75.6±8.1|**72.4**±3.6|**62.5**±9.9|28.3±7.1|

---

> ### Author Response · Authors · 2022-08-02
> **Author Response to Reviewer LUHK:**
>
> We thank the reviewer for the constructive comment. The responses are listed below:
>
> **Q1: My main concern is the performance of the proposed method as compared to existing baselines and FAGCN.**
>
> A1: Compared to the existing baselines, although our NFGNN achieves a better performance on the whole, the performance gains in Table 1 are not adequately significant as expected. In fact, for the 10 popularly referred benchmark datasets in GNN evaluation, we have witnessed many SOTA works that have been verified on them, which makes large performance improvement on them very challenging. Even so, among 10 datasets for evaluation, NFGNN achieves the best on 7 datasets and second-best or comparable results on 3 other datasets. Therefore, we think that NFGNN is competitive with existing baselines.
>
> FAGCN also employs frequency information and gains good performance. However, due to the different settings of dataset construction and splitting, the results reported in the original paper of FAGCN cannot be compared directly with NFGNN.  As you suggested, under the same experimental setup, we re-run FAGCN using the available source code. We also added the comparison with FAGCN in Table 1 of the revised version ( and the following Table R1-1). As we can see, our NFGNN completely outperforms FAGCN on 9 datasets in addition to Cora.
>
> **Q2: Additional experiments on graph classification tasks will further strengthen the paper.**
>
> A2: It is a valuable suggestion that was also pointed out by Reviewer 3. Although we are mainly oriented towards node classification, additional graph classification experiments could indeed help to evaluate our model more comprehensively.
>
> To verify the performance of NFGNN on full graph classification, the TU datasets, the widely used graph classification benchmarks, are adopted. We strictly follow the experimental setup in [1] and also use the same baselines as in it, including GCN, GraphSAGE, GIN, and GAT. Meanwhile, we also choose additional GPRGNN and BernNet for comparison on the graph classification task. As shown in Table R1-2, our NFGNN is also competitive on the graph classification task and outperforms the other baselines on 3 datasets. Particularly, NFGNN achieves the best results among spectral-based methods like GCN, GPRGNN, BernNet, and NFGNN on all datasets.
>
> **Q3&Q4: Even though Sect. 6 is well motivated, the content does not justify it to be a separate section. Citation method is not proper.**
>
> A3: Thanks for your thoughtful comments. We have integrated Sect.6 and Sect.7 into one section to improve the completeness of the description. Besides, more analysis on the connection of existing GNNs with our NFGNN from a polynomial filtering perspective is provided in the supplemental material A.3.
>
> A4: We have changed the citation to the correct format in the revised version.
>
> **Q5: The purpose of reparameterization with low-rank approximation against polynomial.**
>
> A5: We are sorry for the unclear description of it. The purpose of the reparameterization of the localized parameter matrix $\Psi$ of the polynomial via the low-rank approximation can be as follows:
>
> 1. It can significantly reduce the parameter complexity from O(n×(K+1)) to O(d×(K+1+f)). Also, it also provides a way to flexibly adjust the capacity of the model by varying the rank d.
>
> 2. Using matrices $\mathbf{H}$ and $\Gamma$ to approximate $\Psi$ explicitly establishes a bridge between the node-oriented localized filtering and global-shared filtering. If we only use the node-agnostic matrix $\Gamma$, then NFGNN will simply become global-shared filtering.
>
> 3. As we can see from Eq.(9), if we learn $\Psi$ directly in real implementation, only the gradients from $z_i$ will be used to update the parameters of the localized filter corresponding to node i, i.e., $\Psi_{i\ :}$. In other words, only $x_i$ will participate in the optimization of $\Psi_{i\ :}$, which will inevitably lead to an inefficient optimization for $\Psi$. The re-parameterization trick allows us to elegantly solve this issue of optimizing $\Psi$.
>
> **Q6: How does generalized translated operator $\mathbf{T}_i$ help in adapting to local patterns? What is the intuition of $\mathbf{T}_i$ in spatial domain?**
>
> A6. As pointed out in [2], a K-order polynomial spectral filter is strictly localized $N_{<K}(i)$ of node i. Meanwhile, as shown in definition 4.1, $\mathbf{T}\_i$ is with good capability of centering the filter at a specified node, which also means it can help in adapting to local patterns. Specifically, we can learn a set of optimal filter parameters that are specific to node $i$ by locating a K-order polynomial filter in $N_{<K}(i)$ through $\mathbf{T}_i$. In essence, as given in definition 4.1, the $\mathbf{T}_i$ can be intuitively seen as a convolution operator with a delta function centered at node i in the spatial domain[2].
>
> [1] Zhang et al. Nested Graph Neural Networks. NeurIPS2021
>
> [2] Shuman et al. Vertex-frequency analysis on graphs. ACHA2011.

---

### Meta-Review · Area_Chair_RL7W · 2022-08-30

**Recommendation:** Reject
**Confidence:** Less certain

**Metareview:**

The paper has mixed reviews. While some reviewers feel that the paper is novel and interesting, other reviewers think that additional experiments are needed to justify the proposed method and that the proposed methods are somewhat incremental.
The paper will benefit from another revision that will address the raised concerns.

**Award:**

No

---

### Decision · Program_Chairs · 2022-09-14

Reject